# The Interaction between Oxidative Stress Biomarkers and Gut Microbiota in the Antioxidant Effects of Extracts from *Sonchus brachyotus* DC. in Oxazolone-Induced Intestinal Oxidative Stress in Adult Zebrafish

**DOI:** 10.3390/antiox12010192

**Published:** 2023-01-13

**Authors:** Juan Yang, Wei-Wei Zhou, Dong-Dong Shi, Fang-Fang Pan, Wen-Wen Sun, Pei-Long Yang, Xiu-Mei Li

**Affiliations:** Key Laboratory of Feed Biotechnology, Ministry of Agriculture and Rural Affairs, Institute of Feed Research of CAAS, Beijing 100081, China

**Keywords:** *Sonchus brachyotus* DC., extract, oxidative stress, gut microbiota, zebrafish

## Abstract

Oxidative stress is a phenomenon caused by an imbalance between the production and accumulation of reactive oxygen species in cells and tissues that eventually leads to the production of various diseases. Here, we investigated the antioxidant effects of the extract from *Sonchus brachyotus* DC. (SBE) based on the 0.2% oxazolone-induced intestinal oxidative stress model of zebrafish. Compared to the model group, the treatment group alleviated oxazolone-induced intestinal tissue damage and reduced the contents of malondialdehyde, reactive oxygen species, IL-1β, and TNF-α and then increased the contents of superoxide dismutase, glutathione peroxidase, and IL-10. The 16s rDNA gene sequencing findings demonstrated that SBE could increase the relative abundance of *Fusobacteriota*, *Actinobacteriota*, and *Firmicutes* and decrease the relative abundance of *Proteobacteria*. Based on the correlation analysis between the oxidative stress biomarkers and intestinal flora, we found that the trends of oxidative stress biomarkers were significantly correlated with intestinal microorganisms, especially at the genus level. The correlations of MDA, IL-1β, and TNF-α were significantly negative with *Shewanella*, while SOD, GSH-Px, and IL-10 were significantly positive with *Cetobacterium*, *Gemmobacter*, and *Flavobacterium*. Consequently, we concluded that the antioxidant effect of SBE was realized through the interaction between oxidative stress biomarkers and gut microbiota.

## 1. Introduction

It is well known that when intracellular reactive oxygen species (ROS, including O_2_^•−^, •OH, H_2_O_2_, O_3_, and so on) or reactive nitrogen species (RNS, including NO, ONOO^−^, HOONO, and so on) levels exceed the intracellular antioxidant capacity [1,2], it leads to oxidative stress, in severe cases further causing tissue damage, oxidative damage to DNA, lipid peroxidation, and oxidation of protein [3,4,5]. Mild oxidative stress is regulated by the body’s antioxidant system, including superoxide dismutase (SOD), glutathione peroxidase (GSH-Px), catalase (CAT), etc., which is the body’s first line of defense against oxidative stress [6].

Most of the time, the gastrointestinal tract is one of the major sites in living organisms that has an adequate response to oxidative stress due to the generation of pro-oxidants [7], whose production is primarily due to the presence of a plethora of food ingredients, interactions between immune cells, and microbes [1,7]. However, under the production of exacerbating ROS or RNS, these defenses are inadequate and promote the development of intestinal pathology [1,8]. Gut microbiota plays many physiological roles in the organism, including digestion, metabolism, nutrient absorption, vitamin synthesis, prevention of pathogen invasion, and immune regulation [9]. Under normal conditions, the intestinal flora and the host are mutually beneficial and in dynamic equilibrium, but when the balance is disturbed by the influence of external factors, intestinal damage is caused [10,11]. Numerous studies have proven that natural plants and their extracts could significantly adjust the dysbiosis of intestinal flora, promote the growth of beneficial bacteria, inhibit the overgrowth of harmful bacteria, and balance the number of probiotic and pathogenic bacteria, thus keeping the intestinal environment in a healthy state [12,13,14].

*Sonchus brachyotus* DC. is an annual herb of the Sonchus genus in the Compositae family, which can be found in China, Japan, Mongolia, and Russia’s far east. *Sonchus brachyotus* DC. is considered a vegetable and rarely considered to be medicinal, but in fact, *Sonchus brachyotus* DC. is often used as a kind of traditional folk herb to clear heat toxins and stop bleeding and treat acute dysentery, enteritis, and other diseases [15,16]. Although *Sonchus brachyotus* DC. is widely used to treat diseases among people, there are only a few studies and reports on it, especially on its antioxidant effects. A previous study has shown that the methanol extract of *Sonchus brachyotus* DC. has the powers of free radical scavenging and the inhibition of lipid peroxidation [17,18]. Similarly, in our previous research, we demonstrated the ethanol extract of *Sonchus brachyotus* DC. (SBE) obtained with less toxic ethanol as the extraction solvent also has good antioxidant effects, and at the same time, high-performance liquid chromatography (HPLC) analysis was carried out on SBE [15]. However, the antioxidant effect and mechanism of SBE in vivo have not been reported so far.

In this study, we investigated the effects of SBE on oxidative stress biomarkers, intestinal flora, and the interaction between oxidative stress biomarkers and intestinal flora to reveal the antioxidant effect and mechanism of SBE, providing a new reference for the study of the mechanism of antioxidants and laying a foundation for the research and development of SBE as a new antioxidant in aquatic products.

## 2. Materials and Methods

### 2.1. Chemicals and Reagents

Malondialdehyde (MDA), superoxide dismutase (SOD), catalase (CAT), and glutathione peroxidase (GSH-Px) assay kits were all purchased from the Beyotime Institute of Biotechnology (Shanghai, China). The reactive oxygen species (ROS) assay kit was purchased from Nanjing Jiancheng Bioengineering Institute. TRIzol^®^ Reagent was obtained from Thermo Fisher Scientific. Dimethyl sulfoxide and oxazolone were purchased from Sigma-Aldrich (Shanghai, China). EasyScript First-Strand cDNA Synthesis SuperMix was purchased from TransGen Biotech (Beijing, China). Power SYBR Green PCR Master Mix kit was purchased from Abcam (Cambridge, UK). Other chemical reagents used were analytically pure.

### 2.2. Extract Preparation

The aerial parts of *Sonchus brachyotus* DC. were collected from Binzhou City, Shandong Province, China in 2020. At that time, all samples were shaded dry. All dried samples were gained and then passed through a 60-mesh sieve and then ultrasonically extracted according to the conditions of 75% (*v*/*v*) ethanol/water solution, 1:30 (*m*/*v*) material to liquid, and 700 W ultrasonic power for 30 min. Subsequently, after centrifugation at 5000× *g* for 10 min, the eluate was evaporated by rotary evaporation at 40 ± 2 °C to make a powder out of a liquid plant extract using vacuum freeze-drying technology, which was stored at 4 °C [19].

### 2.3. Animal Experiments

Adult wild-type zebrafish were obtained from a commercial dealer (Beijing, China). Twenty zebrafish were housed in 25 L tanks at 28 °C (14/10-h light/dark cycle) and fed a combination of shrimp supplemented with SBE 3 times a day. All zebrafish were treated in compliance with the Ethical Committee for Animal Research of the Institute of Feed Research of CAAS on Experimental Animals (Assurance NO. IFR-CAAS-20210825).

To establish the intestinal oxidative stress model of zebrafish [20] (Table 1), adult zebrafish were randomly divided into 4 groups (30 fish per group, Table 1): control group (CON), ethanol group (EtOH), model group (0.2% oxazolone, Oxa), and treatment group (SBE). The CON group was injected with 0.9% normal saline. The Oxa group was administered an anal injection of 0.2% oxazolone dissolved in 50% ethanol on days 1, 3, and 5, and the EtOH group was injected with 50% ethanol to exclude the effect of ethanol on the oxazolone-induced oxidative stress model. The injection volume for all groups was 0.6 µL/100 mg body weight [20] after 24 h of fasting. Before injection, the belly of the zebrafish was gently squeezed to find the anal opening.

After being injected for 24 h, the zebrafish were treated by feeding shrimp supplemented with 0.3% SBE for 14 days. Then, zebrafish were executed by the hypothermal shock method. Subsequently, the foregut and midgut parts of the zebrafish were collected for biomarker detection and approximately 0.5 cm of the exact location of the hindgut was taken for histological analysis. At the same time, fresh feces samples were taken from the bowel of all zebrafish and then transferred into 2 mL sterile EP tubes. These tubes were rapidly snap-frozen in liquid nitrogen and stored at −80 °C.

### 2.4. Measurement of Oxidative Stress Biochemical Markers

ROS, MDA, SOD, CAT, and GSH-Px levels were determined by assay kit. All detection methods were performed according to the kit instructions.

### 2.5. Histological Analysis

Zebrafish intestinal tracts were harvested, washed, and fixed with formalin and embedded in 4% paraformaldehyde. Intestinal sections (5 µm) were stained with hematoxylin and eosin (H&E) [21]. At least 3 zebrafish were used for this experiment.

### 2.6. Detection of Primary Active Substances

The polysaccharide was analyzed employing the method described by Yang et al. (2018) [22]. The results were expressed in mg of glucose equivalents/mg of extract. The polyphenol was analyzed employing the method described by Ren et al. (2018) [23]. The results were expressed in mg of gallic acid equivalents/mg of extract. The alkaloids were analyzed according to the method proposed by Pan et al. (2018) [24]. The results were expressed as mg of ammothamnine equivalents/mg of extract. The assays were performed in triplicate. The final results are converted to percentages and displayed in the table.

### 2.7. RNA Isolation and Real-Time Polymerase Chain Reaction

Intestinal tissues were ground with liquid nitrogen and then the total RNA was extracted with Trizol regent. Subsequently, RNA was isolated using the EasyScript First-Strand cDNA Synthesis SuperMix kit; 1 µg RNA was retroactively transcribed into cDNA and then used for a subsequent real-time quantitative polymerase chain reaction. The household gene of β-actin was used. Real-time PCR was performed for interleukin (IL)-1β, tumor necrosis factor (TNF)-α, and interleukin (IL)-1β. The relative expression levels of different cytokines were determined by △Ct method. Design of Rt-qPCR primer sequences using the NCBI website and Primer 5 and the primer sequences of Rt-qPCR are shown in Table 2. 

### 2.8. 16s rDNA Gene Sequencing

Collection of bacterial DNA from intestinal content, sequencing of PCR amplicons, and gene analysis were all performed by Biomarker Technologies Co., Ltd. (Beijing, China). Based on the PacBio sequencing platform, the marker genes were sequenced by single-molecule real-time sequencing (SMRT Cell) method, followed by CCS (circular consensus sequencing) sequence filtering to obtain Optimization-CCS for OTUs (operational taxonomic units) clustering, species annotation, and abundance analysis. Specific primers with barcodes were synthesized based on 16S full-length primers 27F (5′-AGRGTTTGATYNTGGCTCAG-3′) and 1492R (5′-TASGGHTACCTTGTTASGACTT-3′), PCR amplification was performed and the products were purified, quantified, and homogenized to form a sequencing library (SMRT Bell), and the library was first quality-checked and sequenced by PacBio Sequel. Alpha diversity, beta diversity, and significant species difference analysis, etc. at the phylum and genus levels were analyzed to explore the differences among samples [25,26]. 

### 2.9. Data and Statistical Analysis

All data were represented as mean ± standard error (SD). Statistical significance was defined as *p* < 0.05, *p* < 0.01. One-way analysis of variance (ANOVA) was used to test each variable for differences among three groups with GraphPad Prism 9. Microbial diversity analysis was performed using BMKCloud (www.biocloud.net accessed on 1 December 2022). Correlation analysis was performed using the Pearson algorithm and significant correlations were calculated at *p* ≤ 0.05.

## 3. Results

### 3.1. Oxazolone-Induced Intestinal Oxidative Stress in Zebrafish

To elucidate the potential antioxidant effects of SBE on intestinal oxidative stress, we constructed a model of oxazolone-induced intestinal oxidative stress in adult zebrafish (Table 1). As we all know, ROS and MDA are the main biomarkers of oxidative stress. When oxazolone was continuously injected for 1, 3, and 5 days, compared to the CON group, the ROS and MDA contents were significantly elevated in the intestine of zebrafish exposed to oxazolone for 1 day and 5 days (Figure 1). In addition, in terms of ROS levels, the EtOH group was higher than the CON group but was not significantly elevated compared to the Oxa group. There was no significant difference in MDA levels between the CON and EtOH groups. Therefore, we excluded the influence of ethanol on the oxazolone-induced oxidative stress model. Considering all together, we selected 0.2% oxazolone-induced intestinal oxidative stress in zebrafish for 1 day as the candidate experimental model.

### 3.2. Antioxidant Effects of SBE and Main Active Substances

#### 3.2.1. Effects of SBE on Intestinal Tissue Morphology

We used H&E to monitor the histological alterations in the zebrafish gut (Figure 2). After exposure to oxazolone, acute enteritis of the zebrafish intestine was induced, accompanied by loss of intestinal villi and exposure of the muscular layer. Moreover, degeneration, necrosis, and detachment of mucosal epithelial cells could be observed, and a large number of neutrophilic infiltrates were noted in the lamina propria and muscularis of the mucosa. The intestinal inflammation was resolved by SBE treatment. Results showed that in the SBE group, there was a more intact mucosal epithelium, a decrease in goblet cells, necrosis and breakage of individual intestinal villi, and occasional neutrophilic infiltration in the lamina propria of the mucosa. Compared to the Oxa group, the SBE group reduced the histopathological score (an indicator of the severity of enteritis). No histological changes were observed in the CON group.

#### 3.2.2. Effects of SBE on Biomarkers of Oxidative Stress

ROS and MDA contents were detected in the dissected intestinal tissues to determine the protective impact of SBE against intestinal oxidative stress carried on by oxazolone exposure. When compared to the CON group, the level of ROS and MDA were relatively high after exposure to oxazolone. SBE treatment effectively decreased the ROS and MDA contents as compared to the Oxa group (Figure 3A,B). For determining the antioxidant capacity of SBE against oxidative stress in the intestine, this was measured by examining the intestinal antioxidant enzyme activity (Figure 3C–E). Compared to the Oxa group, the activity of SOD and GSH-Px significantly increased by 56.36% and 52.47% in the SBE group, respectively. While the CAT enzyme activity in the SBE group declined below the level of the control group, it rose in the Oxa group to surpass that of the CON group. According to the findings (Figure 2 and Figure 3), SBE could successfully lessen the severity of acute oxidative stress in zebrafish, which means the positive impact of SBE on oxazolone-induced intestinal oxidative stress.

#### 3.2.3. Main Active Substances of SBE 

When it was determined that SBE has antioxidant effects in vivo, we further investigated which types of main active substances are principally exerting an antioxidant effect in the SBE. We measured the content of polysaccharides, alkaloids, and polyphenols in SBE. The results are shown in Table 3; the main active substances of SBE were polysaccharides, alkaloids, and polyphenols, the contents of which were 50.14%, 20.03%, and 7.27%, respectively.

### 3.3. Effects of SBE on Intestinal Inflammatory Factors under Oxidative Stress

An inflammatory reaction can be triggered by oxidative stress, which is a component of the inflammatory response. IL-1β and TNF-α expression in the intestinal tract of zebrafish exposed to oxazolone were considerably higher than in control animals (Figure 4A,B). We also observed decreased expression of IL-1β and TNF-α in zebrafish treated with SBE. Furthermore, oxazolone injection did not result in the expression of IL-10, which was unique from the SBE-fed zebrafish group’s behavior (Figure 4C).

### 3.4. Effects of SBE on Intestinal Microbiota in Zebrafish

We performed 16S rDNA gene sequencing to investigate whether SBE treatment alters the composition of gut microbiota in zebrafish. The zebrafish injected with oxazolone and treated with SBE indicated variation in the intestinal microbial population. As for the CON, Oxa, and SBE groups, the distinct OTU counts were 9, 30, and 30, respectively (Figure 5A). The Shannon diversity index fell by 19% and rose by 28% in the Oxa and SBE groups, respectively, compared to the CON group, showing that exposure to oxazolone reduced and SBE enhanced the community variety of the zebrafish gut flora (Figure 5B). The PCoA analysis further indicated a distinct separation of the gut microbiota at the family level between the Oxa group and the SBE groups. (Figure 5C). The intestinal flora of zebrafish were dominated by the *Proteobacteria* and *Firmicutes* (Figure 5D) in the CON, Oxa, and SBE groups. In comparison with the CON group, the *Proteobacteria* was significantly reduced (*p* < 0.05), while the *Firmicutes* showed an increasing trend (no significance), in the SBE group. It was found that the SBE group significantly enhanced the relative abundance of *Fusobacteriota* and *Bacteroidota*, which were compared with the CON group. It was further found at the genus level that the Oxa group increased the abundance of *Kinneretia*, *Edwardsiella*, and *Gemmobacter* as well as decreased the abundance of *Aeromonas* compared to the CON and SBE groups.

### 3.5. Effect of the Association between the Biomarkers of Oxidative Stress and Intestinal Microbiota in Zebrafish

To predict a relationship between the biomarkers of oxidative stress and the diversity of gut microbial, an additional correlation analysis was carried out (Figure 6). When the relationships between oxidative stress biomarkers and intestinal microbial abundance were analyzed at the phylum level (Figure 6A), changes in MDA content were negatively connected to *Actinobacteriota* that exhibited a declining tendency after Oxa injection, and a positive significant correlation was found between GSH-Px and *Plancromucetota* and *Fusobacteriota* (Figure 6A). The abundance of intestinal flora at the phylum level with significant correlation with variation in biomarkers is displayed as histograms (Figure 6B–D). The SBE group increased the abundance of *Actinobacteriota*, *Fusobacteriota*, and *Plancromucetota*, particularly *Fusobacteriota*.

Then, in the further analysis of the genus level, it was found that the content of MDA and ROS were negatively correlated with the abundance of *Shinella*, *Aeromonas*, *Shewanella*, and *Rhizobium*, particularly *Shewanella*, and were considerably reduced (Figure 6E,F). Since SOD and GSH were significantly correlated with a variety of different bacteria as opposed to CAT, only the histogram of the abundance of bacteria whose abundance varied significantly between groups is shown (Figure 6E,G–I). The abundance of *Shewanella* significantly decreased after oxazolone treatment. However, the abundance of *Cetobacterium*, *Flavobacterium*, and *Gemmobacter* significantly increased in the SBE group (Figure 6J).

The correlation analysis between the inflammation factors and intestinal microbial abundance revealed that only IL-10 showed a significant positive correlation with *Plancromucetota* and *Fusobacteriota* at the phylum level (Figure 7A–C), which was consistent with the situation of GSH-Px (Figure 6C,D). At the genus level (Figure 7D), the abundance of genera with significant correlation with cytokines was analyzed, which was found to be significant between groups in terms of the abundance of *Shewanella*, *Cetobacterium*, *Gemmobacter*, and *Flavobacterium*. The results in Figure 7 show that SBE treatment caused an increase in the abundance of *Cetobacterium*, *Gemmobacter*, and *Flavobacterium* and significantly regulated the expression of IL-10 for the improvement of inflammation (Figure 7I).

## 4. Discussion

Zebrafish have been widely used as a relatively inexpensive in vivo model for discovering the effects of environmental chemical exposures on various biological processes directly related to health [27,28]. At the same time, the host provides a nutrient-rich environment for its microbes, which serves crucial functions in host health, growth, aiding digestive functions, and protecting against intestinal infections [29]. There are three forms microbiota found in fish intestinal microflora: facultative anaerobic, obligately anaerobic, and aerobic [30]. So, using fully developed zebrafish as a pre-screen model to guide studies in aquaculture species may contribute to elucidating mechanisms between feed and host-microbe interactions [31]. The zebrafish is therefore a useful and powerful model to study host-bacteria interactions in detail [32]. The study of antioxidant activity of SBE in vitro was apparent in our previous study [15], while the antioxidant effect and mechanism of SBE in vivo are unclear. Therefore, we evaluated the antioxidant effect and mechanism of SBE against oxazolone-induced intestinal oxidative stress in terms of the interaction between oxidative stress biomarkers and gut microbiota. 

As we all know, ROS [33,34] and MDA [34,35] are biomarkers of oxidative stress. Our research showed that that oxazolone could successfully cause intestinal damage and exhibit oxidative stress in zebrafish (Figure 1 and Figure 2). Based on the oxazolone-induced intestinal oxidative stress model in zebrafish, the regulatory effect of SBE on oxidative stress biomarkers was evaluated. In contrast, the SBE group significantly reduced the levels of ROS and MDA compared to the Oxa group (Figure 3A,B). In addition, SOD, CAT, and GSH-Px are the major antioxidant enzymes that detoxify free radicals as the first line of antioxidant defense and are important enzymes in the process of resisting oxidative damage [36,37]. In the SBE group, CAT levels were decreased while SOD and GSH-Px levels were significantly increased (Figure 3C–E). Minor oxidative stress can induce antioxidant enzyme activity, whereas severe oxidative stress can inhibit enzyme activity due to oxidative damage and impaired compensatory mechanisms [38]. CAT is an enzyme from the antioxidant system that is responsible for the scavenging of H_2_O_2_, and the induction of CAT activity indicates a higher production of H_2_O_2_ in organisms [39,40]. In UV toxicity studies [41] on zebrafish embryos, it was found that oxidative stress conditions did not induce changes in CAT activity in zebrafish, while Campos et al. found that exposure to 4-MBC inhibited CAT activity, and our results showed the same inhibition of CAT enzyme activity.

A large number of studies have shown that some active ingredients in natural plants have significant antioxidant effects [42], including small-molecule compounds such as polyphenols, vitamins, saponins, alkaloids, and large-molecule products such as polysaccharides and peptides [42,43,44], which, having in vitro reduction and DPPH scavenging abilities, and so on, enhance antioxidant enzyme activity and decrease peroxidation processes in in vivo antioxidant capacity [45,46]. According to the study of structure-function relationships [42,47], the phenolic hydroxyl groups in the structure of polyphenols can act as hydrogen donors to scavenge free radicals, and their reactivity is closely related to the number and position of phenolic hydroxyl groups such that the adjacent phenolic hydroxyl structure easily reacts with free radicals to form quinone, so most polyphenolic compounds have good antioxidant effects in vitro and in vivo. SBE contains many compounds such as polysaccharides, polyphenols, and alkaloids, which have shown antioxidant effects in vitro and in vivo. Presumably, it is the large content of polysaccharides, polyphenols, and alkaloids from SBE that are responsible for its obvious antioxidant activity.

Inflammation triggered by inflammatory cytokines further accelerates oxidative stress [7,48]. In recent years, there is evidence that oxidative stress plays a crucial role in the development of inflammation [49]. Oxidative stress induced by ROS production stimulates the initial inflammatory response through positive feedback, leading to additional ROS production and in turn further tissue damage [50]. Under normal conditions, the balance between proinflammatory (TNF-α, IL-1, IL-6, IL-8, IL-17, and IL-23) and primary anti-inflammatory (IL-5, IL-10, IL-11, and TGF-β) cytokines is tightly controlled in the GI mucosa [34], but in oxidative stress, IL-1β and TNF-α horizontally rise [51,52,53]. In our study, the increased expression of TNF-α and IL-1β demonstrated the enteritis of zebrafish exposed to oxazolone was caused by oxidative stress, and the addition of SBE reduced the relative mRNA expression of inflammatory factors (Figure 4A,B), and some data showed that the expression of IL-10 was enhanced under the effect of antioxidants [37,54,55]. Similarly, in our study, the expression of IL-10 was significantly increased in the SBE group (Figure 4C), indicating that oxazolone-induced enteritis in zebrafish models could be alleviated by suppressing oxidative stress, lowering the expression of pro-inflammatory cytokines, and upregulating the expression of anti-inflammatory cytokines (Figure 2 and Figure 4).

It has been reported that taxa present in the core gut microbiota of adult zebrafish are conserved and dominated by members of the phylum *Proteobacteria*, followed by *Firmicutes* or *Fusobacteria*. In addition, *Actinobacteria* and *Bacteroidetes* are also present but as subdominant populations [28,56,57,58]. According to the literature [37,55,59,60,61], this may lead to an increase in the relative abundance of *Proteobacteria* and *Bacteroidetes* and a decrease in the relative abundance of the *Fusobacteria* and *Firmicutes* in a zebrafish oxidative stress model. In our study, *Proteobacteria* and *Firmicutes* were the dominant bacteria in the CON and Oxa groups, while the abundance of *Firmicutes*, *Actinobacteria*, and *Fusobacteria* increased and *proteobacteria* decreased in the SBE group (Figure 5). 

At present, the analysis of intestinal microbiota mainly focuses on the changes in flora abundance, but the interaction between gut microbiota and biomarkers of oxidative stress is unclear. Therefore, in this work, we conducted a correlation analysis between the abundance of gut microbiota and the content of oxidative stress biomarkers and selected the microbiota not only with significant correlations but also with significant interactions between groups for discussion and analysis (Figure 6 and Figure 7). In the correlation analysis between oxidative stress biomarkers and intestinal microorganisms, the phylum *Actinobacteria* and genus *Shewanella* have significant negative correlations with MDA content. At the phylum level, compared with the CON group, oxazolone injection caused a decrease in the abundance of *Actinobacteria*, while SBE treatment increased the abundance of *Actinobacteria* like the study [62] about the addition of stabilized fermentation products of commensal *Cetobacterium* somerae XMX-1 to the diets of zebrafish. At the genus level, the Oxa group reduced the abundance of *Shewanella,* and SBE treatment started to recover the *Shewanella* abundance, which may be related to the protease encoded by the sigma factor in *Shewanella* [63,64]; *Shewanella* strains have been shown to act as probiotics in fish farming to protect eukaryotes from pathogens [65]. 

Defining the contribution of host genetic and cellular factors to intestinal microbiome assembly is imperative not only to understand the gut-microbe interactions that are essential to the maintenance of intestinal homeostasis but also to provide insight into disorders associated with microbiota alterations, including cancer, diabetes, oxidative stress, and inflammatory disease [28,66,67]. When antioxidant enzyme levels were correlated with gut microorganisms, GSH-Px was found to have a significant positive correlation with *Fusobacteriota* and *Planctomycetota*, with the SBE group significantly increasing the abundance of *Fusobacteriota* at the phylum level. Species abundance analysis at the genus level associated with either IL-10 or GSH-Px showed significant differences between groups for *Shewanella*, *Cetobacterium*, *Gemmobacter*, and *Flavobacterium* were significant between groups. *Cetobacterium* is an acetate producer, belongs to *Fusobacteria*, and contributed to promoting glucose utilization, regulation of glucose homeostasis, and the improvement of intestinal health in zebrafish [68,69]. Zhang et al. found that the nuclease treatment enhances the probiotic effect of *Bacillus velezensis* T23 performance in reducing intestinal injury and inflammation and significantly increasing the abundance of *Cetobacterium* of the phylum *Fusobacteriota*. *Gemmobacter* is considered to contribute to host nutrition and health by providing complementary enzymatic activities and synthesizing vitamins [70,71]. *Gemmobacter* is strongly associated with amino acid metabolites, including gabapentin and D-serine [71]. *Flavobacterium* is closely related to disease and inflammation in fish [72], and the study of the antioxidant activity of Spirulina platensis polysaccharide showed a significant increase in the abundance of *Flavobacterium* [73]. At the same time, we found that SBE group significantly caused an increase in the abundance of *Shewanella*, *Cetobacterium*, *Gemmobacter*, and *Flavobacterium* compared with the Oxa group (Figure 6 and Figure 7). So, we concluded that SBE regulates oxidative stress biomarkers by playing an anti-oxidative stress role via *Shewanella*, *Cetobacterium*, *Gemmobacter*, and *Flavobacterium*.

## 5. Conclusions

In summary, we found that SBE significantly reduced oxazolone-induced intestinal oxidative stress and restored tissue damage in zebrafish. It was found that SBE significantly increased the abundance of *Fusobacterium*, *Firmicutes*, and *Actinobacteriota* while decreasing the level of *Proteobacteria*, suggesting that the SBE could positively influence the microbial populations as well as the gut microenvironment in contrast to oxazolone. The most important thing is that there was a significant interaction between oxidative stress biomarkers and flora abundance at the genus level in explaining the mechanism of the antioxidant effect of SBE. Oxazolone reduced the quantity of the *Shewanella* species, which in turn increased MDA levels and dramatically increased the production of IL-1β and TNF-α. We would venture to speculate that a decreased abundance of *Shewanella* may indicate oxidative-stress-induced peroxidation and inflammation, and we demonstrated for the first time that SBE is beneficial to increasing the abundance of *Cetobacterium*, *Gemmobacter*, and *Flavobacterium* in the oxazolone-induced oxidative stress model to affect the level of GSH-Px, SOD, and IL-10 (which may be related to genes or metabolites of the gut microbes) to prove the antioxidative function of SBE. Therefore, we concluded that the antioxidant effects of SBE were exhibited through the interaction between oxidative stress biomarkers and gut microbiota, providing a new reference for the study of the mechanism of antioxidants and laying a foundation for the research and development of SBE as a new antioxidant in aquatic products.

## Figures and Tables

**Figure 1 antioxidants-12-00192-f001:**
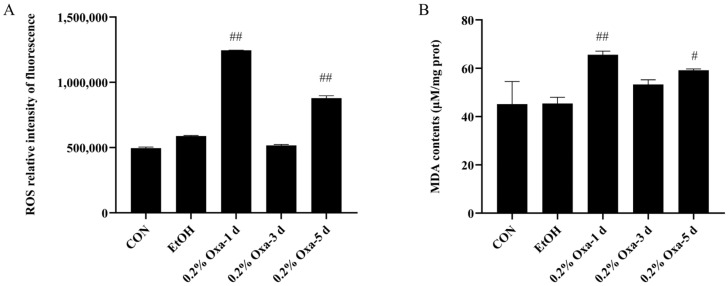
Changes in ROS content and MDA content caused by oxazolone injection time. The values are expressed as the mean ± SD. (n = 6 per group). Statistical differences between groups during the indicated time course were obtained according to repeated-measurement ANOVA (compared with the CON group, ^#^
*p* < 0.05, ^##^
*p* < 0.01).

**Figure 2 antioxidants-12-00192-f002:**
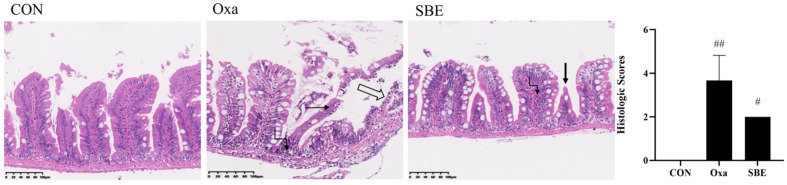
Effects of SBE on the intestinal tissue of zebrafish. Representative hematoxylin and eosin (H&E)-stained intestinal sections and semiquantitative scoring showed tissue damage. The values are expressed as the mean ± SD. (n = 3). The thin black arrows represent degeneration, necrosis, and detachment of mucosal epithelial cells. The hollow arrows represent the exposed muscle layer. The folded arrow represents neutrophil infiltration. Thick black arrows represent villi breakage. Groups with different letters statistically differ (compared with the CON group, ^#^
*p* < 0.05, ^##^
*p* < 0.01).

**Figure 3 antioxidants-12-00192-f003:**
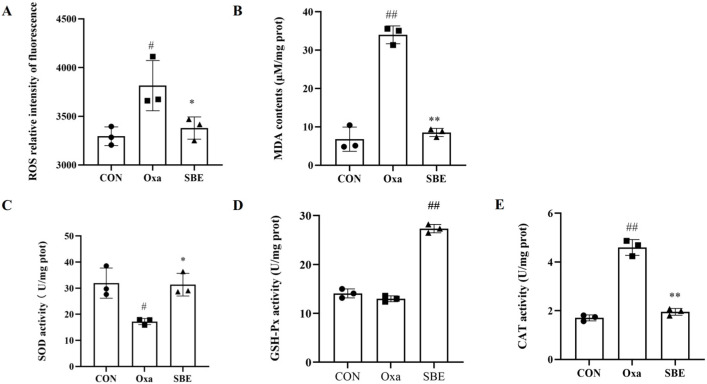
Effects of SBE on the oxidative stress biomarkers. Zebrafish intestines were collected, and the ROS relative intensity of fluorescence (**A**), the MDA contents (**B**), and the enzyme activities of SOD (**C**), GSH-Px (**D**), and CAT (**E**) are shown. Fold changes are expressed as means ± SD (n = 30 per group). Groups with different letters statistically differ (compared with the CON group, ^#^
*p* < 0.05, ^##^
*p* < 0.01; compared with the Oxa group, * *p* < 0.05, ** *p* < 0.01).

**Figure 4 antioxidants-12-00192-f004:**
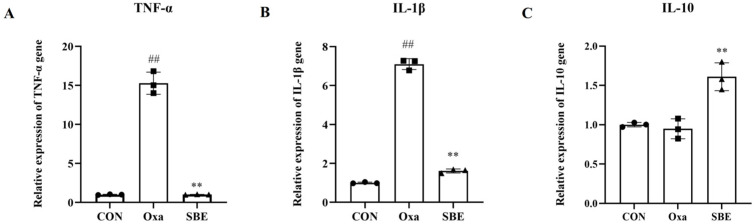
Effects of SBE on the inflammatory factors. Zebrafish intestines were collected, and quantified real-time PCR and mRNA expressions of TNF-α (**A**), IL-1β (**B**), and IL-10 (**C**) are shown. Fold changes are expressed as means ± SD (n = 30 per group). Groups with different letters statistically differ (compared with the CON group, ^##^
*p* < 0.01; compared with the Oxa group, ** *p* < 0.01).

**Figure 5 antioxidants-12-00192-f005:**
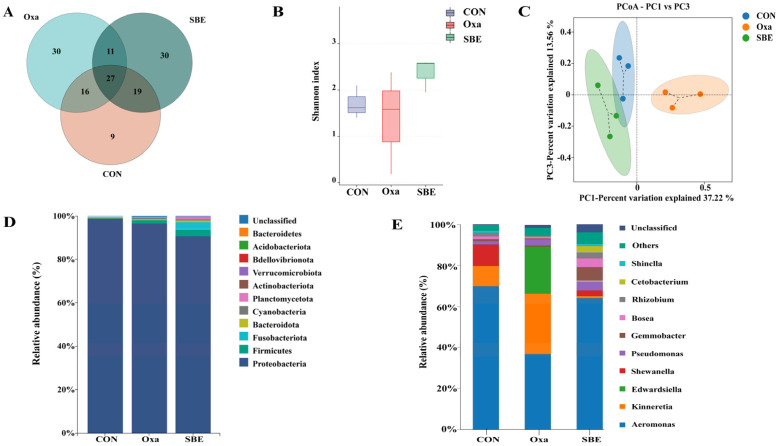
Effects of SBE on intestinal microbiota in zebrafish. A Venn diagram shows the overlap of the OTUs identified in the intestinal microbiota among CON, Oxa, and SBE groups (**A**). Analysis of alpha diversity (Shannon index) was used to detect differences between the groups (**B**). Plots of unweighted UniFrac-based PCoA (**C**). The top 15 bacteria, with a maximum abundance of gut bacteria at the phylum level (**D**). The top 15 bacteria, with a maximum abundance of gut bacteria at the genus level (**E**).

**Figure 6 antioxidants-12-00192-f006:**
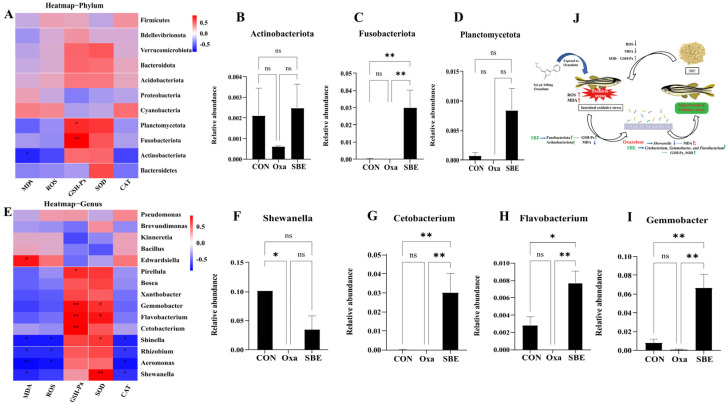
The correlation analysis between the contents of oxidative stress biomarkers and intestinal flora. The variation of oxidative stress biomarkers with relation to the intestinal microbiota. Correlation between biomarkers and intestinal microbiota at the phylum level (**A**). Histogram about the abundance of microbe that correlated with biomarkers contents at the phylum level (**B**–**D**). Correlation between biomarkers and intestinal microbiota at the genus level (**E**). Histogram about the abundance of microbe that correlated with biomarkers contents at the genus level (**F**–**I**). Mechanistic diagram of SBE regulating oxidative stress biomarkers via intestinal flora in zebrafish (**J**). A redder color indicates a positive correlation, while a bluer color indicates a negative correlation (* *p* < 0.05, ** *p* < 0.01. n ≥ 3/per group).

**Figure 7 antioxidants-12-00192-f007:**
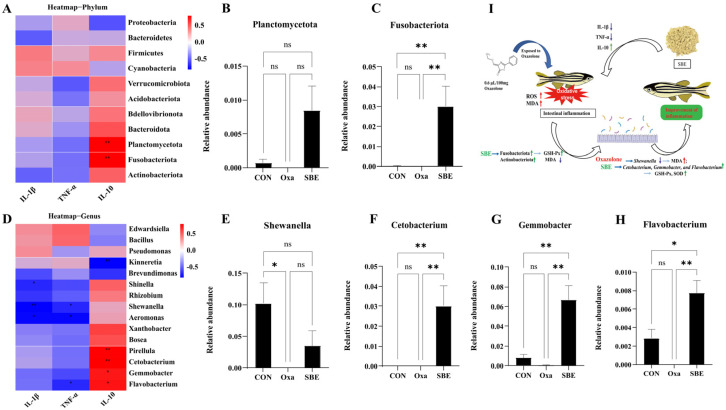
The correlation analysis between intestinal inflammatory factors and intestinal flora. Correlation between the mRNA expressions of TNF-α, IL-1β, and IL-10 and intestinal microbiota at the phylum level (**A**). Histogram about the abundance of microbe that significantly correlated with the expression of IL-10 at the phylum level (**B**,**C**). Correlation between the mRNA expressions of TNF-α, IL-1β, and IL-10 and intestinal microbiota at the genus level (**D**). Histogram about the abundance of microbe that significantly correlated with the expression of TNF-α, IL-1β, and IL-10 at the genus level (**E**–**H**). Mechanistic diagram of SBE regulating inflammatory factors via intestinal flora in zebrafish (**I**). A redder color indicates a positive correlation, while a bluer color indicates a negative correlation (* *p* < 0.05, ** *p* < 0.01. n ≥ 3/per group).

**Table 1 antioxidants-12-00192-t001:** Oxazolone-induced intestinal oxidative stress in adult zebrafish.

Group	0.9% Normal Saline	50% Ethanol	0.2% Oxazolone	0.3% SBE	Quantity per Group
CON	+	−	−	−	30
EtOH	−	+	−	−	30
Oxa	−	+	+	−	30
SBE	−	+	+	+	30

“+” means “added” to the group; “−” means “not added” to the group.

**Table 2 antioxidants-12-00192-t002:** Sequences of RT—qPCR primers.

Gene		Primer Sequence (5′-3′)
*IL-10*	F	TGCGGGCAATATGAAGTC
	R	TTCGCCATGAGCATGTCC
*IL-1β*	F	AGGGCTTTCCTTTAAGACTG
	R	ATATCCCGCTTGAGTTCC
*TNF-α*	F	GGCGCTTTTGGATGTT
	R	TTGCCCTGGGTCTTATG
*β-actin*	F	ACCGCTGCCTCTTCTT
	R	GCAATGCCAGGGTACA

**Table 3 antioxidants-12-00192-t003:** Main active substances in SBE.

Main Active Substances	Contents (%)
Polysaccharides	50.14%
Alkaloids	20.03%
Polyphenols	7.27%

## Data Availability

The dataset supporting the conclusions of this article is available in the Sequence Read Archive (SRA) with the accession code PRJNA894089.

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
