# Peer review of "The Interaction between Oxidative Stress Biomarkers and Gut Microbiota in the Antioxidant Effects of Extracts from Sonchus brachyotus DC. in Oxazolone-Induced Intestinal Oxidative Stress in Adult Zebrafish"

_antioxidants, 2023, doi:10.3390/antiox12010192_

Round 1
Reviewer 1 Report
This is a relatively well written paper on the antioxidative effect on Sonchus brachyotus DC extract. The biological part is well written, although the model is not the most relevant one, but at least cheap. (I think a mammalian model would be better.) My general proplem is still what are the components of SBE extract. The antioxidant molecules (polyphenols/alkaloids?) are not defined. At least an HPLC/MS investigation should be shown in a supplementary material. These nice biological data can not be used for much if we do not know from what componenets of the extract has such positive influence. Ref 16. is hard to identify
Other remarks row 13: delete in the body (are there any diseases outside the body?)
Row 30 the family of ROS is much wider (O3, 1O2, alkyl peroxide) etc.
Ref 4 contains mistype (end of brackets is missing)
Ref 58. Free Rad. Biol Med. Maybe all the abbreviations should have dots like Free. Rad. Biol. Med.
Reviewer 2 Report
The manuscript „Antioxidant effects of the extract from Sonchus brachyotus DC. on oxazolone-induced intestinal oxidative stress to regulate oxidative stress biomarkers via gut microbiota in zebrafish" provides an up-to-date and important information on the antioxidant effects of an extract of Sonchus brachyotus DC. in zebrafish. This data is relevant for a broad public. It is perfectly clear to anyone familiar with kind of researches that it had to be a great effort for authors to plan and conduct such a study, and I have a great appreciation for this. The manuscript is written in clear language, background provides sufficient literature review, the methodology is sound, results are explanatory and well-discussed. Overall, a good read.
I fully support the publication of this paper in Atioxidants.
Author Response
Thanks for your great suggestions.
Reviewer 3 Report
The research article entitled “Antioxidant effects of the extract from Sonchus brachyotus DC. On oxazolone-induced intestinal oxidative stress to regulate oxidative stress biomarkers via gut microbiota in zebrafish” by Juan and colleagues is a valuable contribution to the field of antioxidants. I believe that this manuscript deserves publishing. However, some issues should be previously addressed.
My major concerns regard methodology and syntactical errors (which are found in inappropriate extent) in the text. All my comments are found in detail in the attached text.
I eagerly wait for the revised version of the text.

Author Response
Thanks for your great suggestions. Please see the attachment.

Round 2
Reviewer 1 Report
The authors corrected the manuscript as it was requested, so accept it as it is.
Reviewer 3 Report
The authors have adequately addressed all my comments. Therefore, I suggest publication of this manuscript it its present form.